# Comparative Genomics and Physiological Characterization of Two Aerobic Spore Formers Isolated from Human Ileal Samples

**DOI:** 10.3390/ijms232314946

**Published:** 2022-11-29

**Authors:** Anella Saggese, Rosa Giglio, Nicola D’Anzi, Loredana Baccigalupi, Ezio Ricca

**Affiliations:** 1Department of Biology, Federico II University of Naples, 80125 Naples, Italy; 2Gruppo Savio, Pomezia, 00071 Rome, Italy; 3Department of Molecular Medicine and Medical Biotechnology, Federico II University of Naples, 80131 Naples, Italy

**Keywords:** *Bacillus subtilis*, *Alkalihalobacillus clausii*, antibiotic resistance, probiotics, beneficial bacteria, antimicrobials

## Abstract

Spore formers are ubiquitous microorganisms commonly isolated from most environments, including the gastro-intestinal tract (GIT) of insects and animals. Spores ingested as food and water contaminants safely transit the stomach and reach the intestine, where some of them germinate and temporarily colonize that niche. In the lower part of the GIT, they re-sporulate and leave the body as spores, therefore passing through their entire life cycle in the animal body. In the intestine, both un-germinated spores and germination-derived cells interact with intestinal and immune cells and have health-beneficial effects, which include the production of useful compounds, protection against pathogenic microorganisms, contribution to the development of an efficient immune system and modulation of the gut microbial composition. We report a genomic and physiological characterization of SF106 and SF174, two aerobic spore former strains previously isolated from ileal biopsies of healthy human volunteers. SF106 and SF174 belong respectively to the *B. subtilis* and *Alkalihalobacillus clausii* (formerly *Bacillus clausii*) species, are unable to produce toxins or other metabolites with cytotoxic activity against cultured human cells, efficiently bind mucin and human epithelial cells in vitro and produce molecules with antimicrobial and antibiofilm activities.

## 1. Introduction

Spore formers are mainly Gram-positive bacteria belonging to the *Bacillus* or *Clostridium* genera and sharing the ability to form highly stable and metabolically quiescent spores in response to harsh environmental conditions [1]. Spores are ubiquitous in nature and can be readily isolated from both common and extreme environments, where they can survive indefinitely in the absence of water and nutrients [2,3]. However, in response to the renewed presence of nutrients and favorable environmental conditions, spores germinate, originating vegetative cells able to grow and eventually to sporulate again [1]. Aerobic spore formers of the *Bacillus* genus have long been considered soil bacteria and, indeed, are abundant inhabitants of wet and dry soils and of plant rhizospheres. However, they are also abundantly present in the gut of insects and animals, where they enter as spores with food and water [4,5,6]. In mammals, spores safely transit the stomach and some of them germinate in the intestine, favored by the mildly acidic pH and by the abundance of nutrients [7]. Germination-derived cells grow in the intestine, colonizing that peculiar niche for variable times, before re-sporulating in the terminal part of the intestine and leaving the body in spore form [8,9]. In the animal gut, both spores and germination-derived cells of various *Bacillus* species promote the development of the Gut-Associated Lymphoid Tissue (GALT) [10], ameliorate the effects of oxidative stress and of colitis symptoms [11]. Challenge in vivo studies have also shown that the oral administration of *Bacillus* spores reduces the effects of bacterial [12] and viral infections [13]. In addition, *Bacillus* spore formers have also been shown to modulate the microbial composition of the gut, favoring the prevalence of potentially beneficial bacteria such as *Faecalibacterium prausnitzii* [14], *Akkermansia muciniphila* and *Bifidobacterium* spp. [15]. Spores have been shown to directly interact with immune cells, promoting lymphocyte proliferation within Peyer’s patches and the production of cytokines in mesenteric lymph nodes (MLN) (IL-1, IL-5, IL-6, IFN-ɣ and TNF- α) and in the spleen (IFN-ɣ and TNF-α) [16]. In addition, spores directly protect human cells from oxidative stress by inducing the nuclear translocation of the transcriptional factor Nrf-2 that, in turn, activates stress-response genes [17]. Germination-derived cells of several *Bacillus* species also interact with immune cells contributing to the GALT maturation and to the development of the pre-immune antibody repertoire [10], upregulating the expression of the Toll-like receptors TLR2 and TLR4 [14] and by secreting useful metabolites. Examples of such metabolites are peptides and lipopeptides with antimicrobial activity [18,19] and quorum-sensing peptides, such as the Competence and Sporulation Factor (CSF) of *B. subtilis* [20]. CSFs of *B. subtilis* and CSF-like peptides produced by other *Bacillus* species induce the synthesis of the heat-shock (HS) proteins in human cells in vitro [20,21] and prevent oxidant-induced intestinal epithelial injuries and loss of barrier function, exerting cytoprotective activities in vivo [20].

In this context, we have previously isolated and partially characterized several *Bacillus* strains from ileal biopsies of human healthy volunteers [22]. We now report the genomic and physiological characterization of two of these strains, SF106 and SF174. Both strains were previously shown to be unable to produce the potentially toxic surfactin, to be able to grow anaerobically [22] and to be able to secrete high levels of the cytoprotective peptide CSF (SF106) or CSF-like (SF174) [21]. Such properties, suggestive of a probiotic potential of the two strains, induced their further genomic and physiological characterization. SF106 and SF174 belong to the *B. subtilis* and *Alkalihalobacillus clausii* (formerly *Bacillus clausii*) [23] species, respectively, do not pose safety concerns and produce potentially beneficial molecules.

## 2. Results

### 2.1. Genome Sequence and Phylogenetic Analysis 

Whole genome sequences of SF106 and SF174 were obtained with a coverage of ~30×, with 11 and 40 contigs for SF106 and SF174, respectively (Table 1). A total of 3992 and 4225 CDS, 8 and 12 rRNA and 63 and 75 tRNA-coding genes were assigned for SF106 and SF174, respectively (Table 1). The obtained genomes are approximately 4.0 Mbp long, with 43.75 and 44.67% GC content for SF106 and SF174 (Figure 1 and Table 1). No plasmid sequences were detected in either strain (not shown). Based on the analysis of the sequence of the 16S RNA gene, SF106 and SF174 were tentatively considered as belonging to the *B. subtilis* and *B. clausii* species, respectively [22]. *B. clausii* has recently been re-classified as a new genus, *Alkalihalobacillus clausii* [23] and therein indicated according to the new taxonomy. The Average Nucleotide Identity (ANI) of the two genomes was determined against the genome of a type strain of each species. SF106 showed an ANI of 99.98% with the genome of *B. subtilis* 168, while SF174 had an ANI value of 99.96% with *A. clausii* ENTPro.

The phylogeny of SF106 and SF174 was further analyzed by comparing their genomes with those of several other strains of the same species. Phylogenetic trees, obtained as described in Materials and Methods, clearly indicated that SF106 and SF174 clustered with *B. subtilis* and *A. clausii*, respectively (Figure 2).

#### Functional Annotation of SF106 and SF174 Genomes

The two genomes were analyzed by using the eggNOG database, a public resource classification database able to provide Cluster Orthologous Groups (COGs) of proteins with functional annotations. Each protein of SF106 and SF174 was assigned a COG number and subjected to functional clustering analysis according to the classification criteria of the eggNOG database (Figure 3). Of the 3992 ORFs of SF106 (Table 1), 3950 (93.49%) had COG numbers, while of the 4225 ORFs of SF174 (Table 1), 3950 (94,07%) had COG numbers (Figure 3). In both strains, the majority of the annotated genes were classified into “Function unknown” (S category in Figure 3), representing 24,96% (986 proteins in SF106) and 22,56% (896 proteins in SF174) of the total proteins. According to this annotation system, the next most represented classes were “amino acid transport and metabolism”, “carbohydrate transport and metabolism” and “transcription” (respectively E, G and K categories in Figure 3) in both strains. These categories, as well as those less represented, were overall similarly distributed in the two strains (Figure 3).

Functional annotation of the two genomes was also performed by the BlastKOALA tool on the KO (KEGG Orthology) database, an automatic annotation server for genome sequences, which perform KO assignments to characterize individual gene functions and reconstruct KEGG pathways. Furthermore, in this analysis, which uses different functional categories with respect to eggNOG, many genes of both strains were classified to the Function “unknown category”. A broad function was assigned to the remaining genes, with “genetic information processing”, “signaling and cellular processing”, “carbohydrate metabolism”, “environmental information processing”, “metabolism” and “amino acid metabolism” being the most represented categories (Figure 4). Using this different database, a similar distribution of functional categories was also observed in the two strains (Figure 4).

The SF106 and SF174 genomes were further analyzed by using the CAZy database, a specialized database focused on the analysis of carbohydrate-active enzymes (CAZymes). The analysis revealed a total of 135 and 111 CAZymes for SF106 and SF174, respectively (Table 2). These numbers are similar to those previously observed in other strains of the same species [24] and to those present in selected probiotic strains of the respective species (Table 2). Analysis of the CAZy database revealed the presence of candidate enzymes involved in breakdown of oligo- and disaccharides. In particular, SF106 and SF174 code for candidate beta-d-glucosidase of the glycoside hydrolase 3 (GH3) family and β-galactosidase of the glycoside hydrolase 2 (GH2) family, both proposed to target host glycans as well as bacterial cell walls [25] (Table 3). Both SF106 and SF174 code for a candidate glycoside hydrolase of the GH18 family, proposed to act against animal glycans [26]. (Table 3). Moreover, both genomes have six candidate N-acetylglucosamine deacetylases of the carbohydrate esterase 4 (CE4) family and one N-acetylglucosaminidase-6P-deacetylase of the carbohydrate esterase 9 (CE9) family, proposed to catalyze the elimination of an acetyl group from peptidoglycan N-acetylglucosamine as well as from animal glycan containing O-acetylated sugars (for example, sialic acids) [25] (Table 3).

### 2.2. Genome Analyses

Pangenome analysis was performed by using the Anvi’o platform (https://merenlab.org/2016/11/08/pangenomics-v2; accessed on 1 April 2022). SF106 and SF174 genomes were compared to those of 14 strains of *B. subtilis* or 14 strains of *A. clausii*, selected as close relatives of SF106 and SF174, respectively (Figure 5). The pangenome of SF106 and the other *B. subtilis* strains comprised a total of 3852 genes as gene cluster, with 96.05% as core genes (3700 genes) and 3.94% as accessory genes (only 152 genes) while no unique genes were found (Figure 5A). The pangenome of SF174 and the other *A. clausii* strains comprised a total of 3967 genes as gene cluster, with 87,85% as core genes (3485 genes), 12,15% as accessory genes (only 482 genes) and no unique genes (Figure 5B). A representation of the accessory gene distribution of SF106 and SF174 in bins at least 200 bp in length was performed using the ClustAGE tool [27] (Figure 6). While in the analyzed genomes of *B. subtilis* the accessory genes were carried by large DNA fragments, in *A. clausii* they were mainly distributed in very short bins (Figure 6). In both SF106 and SF174 the accessory genes were at least in part shared by several of the analyzed genomes (Figure 6).

SF106 and SF174 genomes were analyzed for the presence of homologs of genes coding for vitamins, prophages, mobile elements, toxins and antimicrobials. Both genomes potentially code for enzymes involved in the synthesis of vitamin B1, B2, B5, B6, B9 and lipoic acid, while only SF106 also codes for the vitamins B8 and K2 (Appendix A). For vitamin B1, SF106 contains both a pathway for the de novo synthesis and a salvage pathway, while SF174, like all other *A. clausii* strains, is unable to perform the de novo synthesis and only has genes for the salvage pathway. The genes of SF106 or SF174 coding for the enzymes involved in the synthesis of all vitamins indicated in Appendix A are part of the core genome and share a similar organization with those previously found in other members of the respective species (Appendix A).

Phage-like elements, common to many bacterial genomes, were also identified in SF106 and SF174. While in the genome of the *B. subtilis* type strain 168, four phage-like elements are present (indicated as PBSX, PBSY, PBSW and PBSZ), the genome of SF106 contains only two phage-like elements, both with high homologies with PBSX of the 168 strain. Locus SF106_1680-1723 is composed of 43 ORFs and is apparently complete, while the SF106_0483-0518 locus is composed of only 35 ORFs and is therefore incomplete (Appendix A). In the SF174 genome, two phage-like elements are present, one apparently complete (locus SF174_4112-4129) and one defective (SF174_0758-0777), as in the reference genome *A. clausii* ENTPro (Appendix A).

No insertion sequences (IS) have been found in the SF106 genome, while one IS element is present in SF174. The IS of SF174 is identical to that of other *A. clausii* strains, including the commercial probiotic strain ENTPro, with the transposase (SF174_0688) showing a 100% similarity with that (WP_035201381.1) of *A. clausii* ENTPro.

SF106 and SF174 genomes were screened for the presence of genes coding for toxins present in various strains of *B. cereus* but also found in other species of the *Bacillus* genus [28,29]. Homologs of *hblABCD*, *nheABC*, *cytK* and *bceT* genes of *B. cereus* were not detected in either the SF106 or SF174 genome (not shown). Genes putatively coding for hemolysins were also not identified in both genomes (not shown). Accordingly, both strains were found unable to oxidize iron in hemoglobin molecules within red blood cells (alpha-hemolysis) or to lyse red blood cells (beta-hemolysis) and were therefore considered as not hemolytic (gamma-hemolysis) (not shown).

Homologs of genes coding for antimicrobials were identified in both the SF106 and Sf174 genomes. In particular, SF106 codes for enzymes involved in the biosynthesis of three bacteriocins, Subtilosin A, Bacilysin and Kanosamin. Subtilosin A is a cyclic, class I lantibiotic synthesized by ribosomal protein synthesis [30], active against *Listeria monocytogenes* [31] and also able to prevent biofilm formation by some bacteria [32]. In SF106, nine genes coding for the various enzymes involved in Subtilosin A production and secretion are clustered, similarly to what was reported for other members of the *B. subtilis* species (Appendix A, panel A). Bacilysin is a dipeptide produced by several species of the *B. subtilis* group by non-ribosomal peptide synthases (NRPS) [33] and active against *Staphylococcus aureus* [34]. In SF106, the specific NRPS for Bacilysin production and maturation are clustered, as in other members of the *B. subtilis* species (Appendix A, panel B). Kanosamine is an amino sugar with antifungal activity synthesized in various *Bacillus* species by the action of three enzymes, NtdC, NtdA and NtdC [35], encoded by the *ntdABC* operon in *B. subtilis* or *kabABC* in *B. cereus* [36]. In SF106 there are five genes putatively involved in kanosamine synthesis (Appendix A, panel C). SF174 codes for enzymes involved in the biosynthesis of Gallidermin, a 47-amino-acid lantibiotic active against several Gram-positives [37] and for the proteins involved in its maturation and secretion (Appendix A, panel D).

### 2.3. Antimicrobial and Antibiofilm Activities

To validate the genomic predictions on the potential production of antimicrobials, SF106 and SF174 were tested against a panel of target microorganisms by both agar diffusion assay with cell-free supernatants and by cell–cell contact on plates. SF106 supernatant was active against *B. cereus* and *Listeria monocytogenes* (Table 4), probably because of the production and secretion of Subtilosin A. SF106 was also able to inhibit the growth of various *Staphylococcus aureus* strains, including multi-drug and methicillin-resistant (MRSA) strains, by direct cell–cell interaction (Table 4). In addition, SF174 was active against *L. monocytogenes,* but only by cell–cell contact (Table 4), a mechanism of anti-microbial activity that may be due to a low-level secretion of the anti-microbial or to the production of membrane vesicles [38] or nanotubes [39]. Further ad hoc investigations will be needed to fully clarify this point. In addition, cell-free supernatants of both strains were active against a reference strain of *Candida albicans*, activity that, for SF106, could be due to the reported antifungal properties of kanosamine [35]. The results presented in Table 4 are in good agreement with the bioinformatic prediction for SF106, while for SF174, the expected activity against a wide range of gram-positives was not observed, suggesting that the identified genes coding for Gallidermin are either poorly or not expressed in the experimental conditions used.

In addition, SF106 and SF174 cells in the stationary phase of growth secreted molecule(s) able to efficiently inhibit biofilm formation by *B. licheniformis* (only SF106) and *Mycobacterium smegmatis* (both strains) (Figure 7A). Cell-free supernatants of both strains were also evaluated for their ability to dis-assemble pre-formed biofilms and only strain SF174 was active against biofilm pre-formed by *M. smegmatis* (Figure 7B). Since the genomic analysis indicated that SF106 codes for Bacilysin (see above), which is known to have anti-biofilm activity [33], it is likely that the activity of Figure 7A is due to that molecule. The responsible genes for the anti-biofilm activity of SF174 have not been identified by the bioinformatic analysis.

### 2.4. Antibiotic-Resistance

The minimal inhibitory concentrations (MIC) of a panel of eight antibiotics identified as relevant for members of the *Bacillus* genus [40] was determined. As reported in Table 5, SF106 showed MIC values lower than the breakpoints for seven of the eight antibiotics. Strain SF174 showed MIC values lower than the breakpoints for *Bacillus* for four of the eight antibiotics indicated by EFSA and demonstrated resistance to streptomycin, clindamycin, erythromycin and chloramphenicol. For two other antibiotics, kanamycin and tetracycline, MIC values very close to the EFSA breakpoints were shown (Table 5).

Resistance to streptomycin of SF106 was most likely due to the presence of a homolog of the *aadK* gene, encoding an aminoglycoside 6-adenylyltransferase (SF106_0428) known to cause resistance to the antibiotic and common to many *B. subtilis* strains [41,42]. The pangenomic analysis reported above indicated that the *aadK* gene of SF106 is part of the core genome and was carried by all genomes analyzed (Appendix A). Moreover, the analysis of DNA sequences flanking the *aadK* gene (upstream and downstream of the coding part) did not reveal the presence of a repeated sequence that could suggest a possible mobilization of the gene (not shown). In addition, SF106 cells were grown continuously for seven days in LB medium in aerobic conditions at 37 °C, refreshing the culture when it reached the stationary phase of growth (approx. 2.0 OD600 nm). Over 5000 cells were then tested on plate and none of them had lost the streptomycin-resistant phenotype, suggesting that the *aadK* gene was not easily transferable (not shown). Other low-level resistances of SF106, below the EFSA breakpoint values (vancomycin, tetracycline, Table 5), could be due to unspecific efflux pumps also identified in the SF106 genome and can be considered intrinsic resistances [43].

In the SF174 genome, more than a single gene for each analyzed antibiotic was identified as potentially responsible of the phenotypic resistance (Appendix A). A total of 15 genes were identified as potentially coding for the observed antibiotic resistance. Eleven of them were in the core genome and were shared by all *A. clausii* analyzed as reported in the Appendix A, while the remaining four genes were in the accessory part of pangenome and present in 9 (chloramphenicol and streptomycin) or 12 (chloramphenicol and erythromycin) out of 14 genomes analyzed. All 15 antibiotic-resistance genes were also present, with the same genomic organization, in the ENTPro genome, an assembled genome of the four different *A. clausii* strains present in a commercial probiotic for human use (Enterogermina, Sanofi). In addition, the analysis of DNA sequences flanking the various resistance genes (upstream and downstream of the coding parts) did not reveal the presence of known mobile genetic elements or any other repeated sequence that could suggest a possible mobilization of the gene (not shown).

### 2.5. Interactions of SF106 and SF174 with Mucin and Human Epithelial Cells In Vitro

The ability of SF106 and SF174 cells to interact with the intestinal environment was evaluated in vitro by assessing the presence of cytotoxic activities and by measuring the adhesion to human colonic mucin (HCM) and to human intestinal cells. Assessment of cytotoxic activities was performed in vitro by a colorimetric (MTT) assay on HCT116 cells. HCT116 is a human colon cancer cell line characterized by an epithelial morphology that has been widely used to assess cell proliferation during drug screening [44]. When 5 × 10^3^ HCT116 cells were incubated with the supernatant (20% vol/vol) of SF106 or SF174, no reduction in the intracellular NAD(P)H-dependent oxidoreductase activity was observed with respect to the untreated cells (data not shown). These results indicate that SF106 and SF174 do not interfere with the metabolic activity of human cells and, therefore, that no cytotoxic compounds were produced and secreted by the two strains under these experimental conditions. 

Next, we analyzed the adhesion of spores of SF106 and SF174 to human mucin, a factor considered relevant for intestinal colonization by beneficial bacteria. SF106 adhered to HCM slightly more efficiently than a commercial probiotic strain of *B. subtilis* (HU58, Microbiome Labs), known to have positive effects on antibiotic-induced gut dysbiosis [45] (Figure 8A). SF174 was, instead, less efficient than a commercial probiotic strain of *A. clausii* (Enterogermina, Sanofi) (Figure 8A). The adhesion of *Bacillus* spores to human epithelial cells was analyzed in vitro by using HCT116 cells. For this analysis, 4.5 × 10^5^ HCT116 cells were incubated with 2.5 × 10^7^ spores of SF106 or SF174 for 3 h at 37 °C. In parallel, spores of commercial probiotics of the same species were also tested. As reported in Figure 8B, SF106 spores showed an adhesion to human cells similar to that of spores of the commercial strain HU58. SF174 was, instead, slightly more efficient than a commercial probiotic strain of *A. clausii* (Enterogermina, Sanofi) with an average adherence of about three spores/cell (Figure 8B).

## 3. Discussion

The human isolated strains SF106 and SF174 had been previously proposed to belong respectively to the *B. subtilis* and *A. clausii* species by alignment of their 16S gene sequences [22]. However, a 16S rRNA analysis often cannot provide an unambiguous assignment [46] that has been provided in this study by whole-genome Average Nucleotide Identity (ANI) and the phylogenetic analysis. SF106 and SF174 were then classified as strains of the *B. subtilis* and *A. clausii* species, respectively. This is an important point, since both these two species are in the qualified presumption of safety (QPS) list of the EFSA (https://www.efsa.europa.eu/en/topics/topic/qualified-presumption-safety-qps, accessed on 1 April 2022) and several strains belonging to them are widely marketed as probiotics for human and animal use [4,6].

SF106 and SF174 genomes were submitted to a comparative genomic analysis with genomes of close relatives of the respective species. Both genomes do not contain unique genes and only a limited number of accessory genes (genes present in at least one other analyzed strain). Both genomes do not contain genes for known toxins and putatively encode for vitamins and antimicrobials. 

The production of vitamins is considered a positive trait for a probiotic strain. Indeed, depletion of vitamins has been associated to a variety of human diseases, including neurologic and cardiovascular disorders (vitamin B1), weakness (vitamin B2), sensitivity to insulin (vitamin B5), immunodepression (vitamin B6), skin problems (vitamin B8) [47], anemia (vitamin B9) [48], blood coagulation (vitamin K2) [49] and epilepsy (lipoic acid) [50]. Vitamins produced by commensal bacteria may contribute to human nutrition and metabolism. Both SF106 and SF174 contain genes coding for enzymes involved in the synthesis of vitamins B1, B2, B5, B6, B9 and lipoic acid. Only SF106 also encodes enzymes for the production of the vitamins B8 and K2, with the latter known to act as an anticoagulant and recently associated with calcium turnover in the human body [49].

In addition, the production and secretion of molecules with antimicrobial and/or antibiofilm activity is a positive trait for a probiotic bacterium, and could be used to inhibit the growth of potentially pathogenic bacteria. In this context, it is noteworthy that both strains have antimicrobial, antimycotic and antibiofilm activities. For SF174 the bioinformatic analysis did not reveal possible responsible gene(s) for the experimentally observed antibiofilm activity, suggesting that it could be due to a novel molecule produced by that strain.

Both strains are resistant to some antibiotics; SF106 is resistant to streptomycin and SF174 to streptomycin, clindamycin, erythromycin and chloramphenicol, and weakly to kanamycin and tetracycline. Resistance of SF106 to streptomycin, due to the presence of a homolog of the *aadK* gene, is common to many *B. subtilis* strains [41,42]. Indeed, the *aadK* gene is part of the core genome in the pan-genomic analysis reported in this study and is not easily transferable. Resistance of SF174 could be due to fifteen genes found to be homologous to previously identified resistance genes. Eleven of these are in the core genome, while the remaining four genes are in the accessory genome and are shared by several other strains (two genes are present in 9 out of 14 genomes and the other two in 12 out of 14). In this case, the lack of mobile elements in the proximity of these genes also makes their mobilization unlikely. In addition, all fifteen resistance genes are also present in the genome of probiotic strains of *A. clausii* commercialized for over 50 years for human use. Altogether, the wide distribution of these genes and the very low probability of their mobilization point to their presence not being dangerous or even advantageous when probiotic therapy is used in combination with antibiotic therapy.

In the human intestinal tract, in the region between the duodenum and the terminal part of the ileum, the epithelial cells are covered and protected by a thick mucus layer composed mainly of mucin-type glycoproteins. Therefore, efficient adhesion to human colonic mucin (HCM) is considered an additional factor contributing to intestinal colonization by beneficial bacteria. SF106 adhered to HCM slightly better than a commercial probiotic strain of the same species, while SF174 was less efficient than the commercial strain. Both strains were also compared with the respective controls for the ability to adhere to HCT 116 cells, a model of intestinal epithelial cells commonly used for studying the interactions between bacteria and intestinal epithelial cells [48]. SF106 performed similarly to its control, while SF174 was slightly more efficient than the commercial *A. clausii* strain. Both adhesion assays were performed with spores of the two strains (and of their respective controls). The use of spores was based on the fact that all commercial probiotics based on spore formers contain spores, mostly because spores are more stable, have a longer shelf-life and survive better the transit through the stomach than vegetative cells. As a consequence, the spore is the cell form that reaches the intestine during a probiotic treatment and has to adhere to mucin or to the intestinal cell surface as the initial step of the temporary colonization of the intestine.

Altogether, the genomic and physiological characterization of SF106 and SF174 indicate that the two strains do not pose safety concerns and have beneficial potential. Such potential probiotics will need to be confirmed by in vivo trials with animal models that will be a challenging, future research step.

## 4. Materials and Methods

### 4.1. Bacterial Growth, Spore Production and Purification

SF106 and SF174 [22] cells were grown in LB medium (for 1 L: 10 g Bacto-Tryptone, 5 g Bacto-yeast extract, 10 g NaCl, pH 7·0) at 37 °C. For antimicrobial and antibiofilm assays (see below), late stationary cultures (grown about 24 h) were centrifuged (1000× *g* for 10 min at Room Temperature) and the supernatant was filter-sterilized with a 0.22-µm filter (Millipore, Bedford, MA, USA). 

Spores were prepared by the exhaustion method [51] using Difco Sporulation Medium (DSM) (for 1 L: 8 g/L Nutrient Broth, 1 g/L KCl, 1 mM MgSO_4_, 1 mM Ca(NO_3_)_2_, 10 μM MnCl_2_, 1 μM FeSO_4_, Sigma-Aldrich, Germany) and incubated at 37 °C for 30 h. Before purification, spores were washed four times with cold, sterile distilled water, centrifuged at 8000× *g* for 20 min and then purified using gastrografin gradient centrifugation as described by Nicholson et al. [49], followed by resuspension in sterile water. The purified spore suspension contained >90% phase bright spores, as confirmed by observation under the light microscope.

### 4.2. Whole-Genome Sequencing and Bioinformatic Analysis of SF106 and SF174 Genomes

Exponentially growing cells were used to extract chromosomal DNA as previously reported [52]. Genome sequencing of SF106 and SF174 was performed by GenProbio (Parma, Italy) with Illumina MiSeq Sequencing System. Genome assembly was performed with SPAdes v3.14.0 by means of MEGAnnotator pipeline [53]. The complete genome sequences of SF106 and SF174 were used for phylogenetic analyses, including several genomes belonging to *B. subtilis* and *A. clausii* species obtained from the National Center for Biotechnology Information. Two genomes of *B. licheniformis* and *B. cereus* were chosen as an outgroup. The phylogenomic trees produced by GTDB-Tk were visualized on the Interactive Tree of Life (iTOL, platform v6). Average Nucleotide Identity (ANI) values between the sequenced genomes and the closest bacteria were obtained using Ezbiocloud tool (https://www.ezbiocloud.net/tools/ani, accessed on 1 April 2022). The genomes of the SF106 and SF174 have been deposited in GenBank as BioProject PRJNA882357 (accession number CP104962) and PRJNA882359 (accession number CP104963), respectively. 

### 4.3. The Genome Annotations and Genome and Pangenome Analysis

Functional annotations of SF106 and SF174 genomes were obtained from the KEGG database (available at KEGG Web site https://www.kegg.jp/; accessed on 1 April 2022, release 102.0) to identify protein-coding genes and their associated KEGG Orthology (KO) annotations. Similarly, the genomes were also scanned for cluster of orthologous groups (COGs) annotations using eggNOG-mapper v2 (http://eggnog-mapper.embl.de, v2.1.5, accessed on 1 April 2022). 

Secondary metabolite biosynthesis gene clusters were searched for using the web-based genome mining tool antiSMASH (http://antismash.secondarymetabolites.org, accessed on 1 April 2022, version 6.0).

The presence of prophage sequences in the *Bacillus* genomes was analyzed with the PHASTER search system (http://phaster.ca/, accessed on 1 April 2022). The presence of antimicrobials, putative toxins and genes involved in the production of vitamins was predicted using GhostKOALA, KEGG tools for functional characterization of genome. All results obtained were verified using with BLAST with Ubuntu, creating a specific database of protein sequences for each target. 

Functional annotations of the two genomes were also assigned using the dbCAN2 server (https://bcb.unl.edu/dbCAN2/blast.php, accessed on 1 April 2022) that integrated three tools/databases for automated CAZyme annotation: HMMER for annotated CAZyme domain boundaries according to the dbCAN CAZyme domain HMM database; DIAMOND for fast blast hits in the CAZy database; HMMER for dbCAN-sub a database of carbohydrate active enzyme subfamilies for substrate annotation. 

The pangenomes of *Bacillus subtilis* SF106 and *Alkalihalobacillus clausii* SF174 (15 in total for each species: 14 were obtained from the National Center for Biotechnology Information) were obtained with anvi’o (available from http://github.com/meren/anvio, accessed on 1 April 2022, version 7.1), an open-source, community-driven analysis and visualization platform for microbial-omics. Briefly, a database was created with Prodigal (version 2.6.3), used to identify open reading frames in contigs; subsequently, to populate the contigs database with more information, NCBI’s Clusters of Orthologous Groups database was used, which assignees functions to genes, and the KEGG KOfam database, which annotates the genes in the database. The combination of tools Spine and AGEnt was used to define the core and accessory genes (http://vfsmspineagent.fsm.northwestern.edu, accessed on 1 April 2022, version 0.3.1).

### 4.4. Antimicrobial and Antibiofilm Activities

Antimicrobial activity was determined using the method described by Schillinger and Lüeke [54] with the following modifications: 10 µL of each bacterial culture in stationary phase of growth or 10 ul of cell-free supernatants (see section above) were spotted on the surface of a sterile LB agar plate and the spots air-dried. Then, 100 µL of an exponential culture of each of the indicator bacterial strains (indicated in Table 3) was mixed with 10 mL of soft agar (0.7%) and poured over the plate. Fresh media were used as negative controls. The plates were incubated aerobically overnight at 37 °C and the presence and diameter of the inhibition halos were evaluated. 

### 4.5. Antibiofilm Assay 

Biofilm formation was tested by growing bacterial cells in 24-well culture plates containing LB medium for *Bacillus licheniformis* and Sauton medium (0.05% KH_2_PO_4_, 0.05% MgSO_4_7H_2_O, 0.2% citric acid, 0.005% ferric ammonium citrate, 6% glycerol, 0.4% Asparagine, 0.05% Tween 80, pH 7.4) for *Mycobacterium smegmatis* [55] and incubated in a static condition at 37 °C for 48 h and 1 week, respectively. For pre-exposure experiments, cell-free supernatants of SF106 and SF174 were added to the wells together with the bacterial target strains while for post-exposure experiments the supernatant was added on the pre-formed biofilm and incubated for 24 h at 37 °C. Biofilm production was assessed by crystal violet assay as previously described in Vittoria et al. [56]. Data were normalized by total growth as measured by OD590 nm readings, and the experiment was performed in triplicate.

### 4.6. Minimum Inhibitory Concentrations

MIC evaluation of SF106 and SF174 was determined according to the methodology described in the EFSA documentation [40]. The cells were exposed to 10 serial dilutions of each antibiotic in sterile broth. Following an appropriate incubation period, the MIC of each antibiotic was determined. The strains deemed susceptible or resistant to particular antibiotics based on specific MIC thresholds established by the EFSA for *Bacillus* strains are reported in Table 4.

### 4.7. MTT Assay

Cytotoxicity study on human cells was performed by using epithelial (HCT 116) cells that were cultured in McCoy’s 5a medium (Sigma Aldrich, Milan, Italy) supplemented with 10% fetal bovine serum (Hy-Clone, GE Healthcare Lifescience, Chicago, IL) and 1% penicillin-streptomycin, at 37 °C in humidified atmosphere of 5% CO_2_. Cytotoxicity on HCT 116 cells was assessed by performing the MTT reduction inhibition assay. Cells were grown as described and plated on 96-well plates at a density of 5 × 10^3^ cells per well, in 200 µL of medium containing SF106/SF174 cell-free supernatants (20% *v/v*) for 24 h [19]. After treatment, the medium was eliminated and 10 µL of a stock MTT reagent was added into each well to a final concentration of 0.5 mg/mL. After 4 h incubation, the MTT solution was removed and the MTT formazan salts were dissolved in 100 µL of DMSO. Cell survival was expressed as the absorbance of blue formazan measured at 570 nm with an automatic plate reader (Multi scan spectrum; Thermo-Fisher Scientific, Waltham, MA, USA). Cytotoxicity tests were performed at least 3 times and cell survival values expressed as percentage of viable cells with respect to untreated samples. The MTT reagent was prepared by dissolving 5 mg of 3-(4,5-dimethylthiazol-2-yl)-2,5-diphenyl-tetrazolium bromide (MTT) (Sigma-Aldrich) in 1 mL PBS. DMSO was used for solubilization of the formazan crystals.

### 4.8. Mucin Adherence Assay

Ninety-six-well plates were incubated overnight at 4 °C with 0.5 mg/mL of mucin from pork stomach (Sigma-Aldrich, USA) as described elsewhere [57]. Mucin-coated wells were incubated with 40 μL of spore suspensions (5 × 10^6^ spores per well) for 3 h at 37 °C, unbound spores were rinsed off with 3 washes with PBS, while bounded spores were subsequently removed with 100μL of 0.06% Triton X-100 for 30 min at 37 °C and plated onto LB agar plates, incubated aerobically for 24 h at 37 °C, and colony-forming units (CFU) per ml counted. Total spore count was determined from unwashed mucin-coated wells.

### 4.9. Adherence to HCT116 Cells 

HCT 116 cells were grown at 37° C with 5% CO_2_ in McCoy’s 5a medium supplemented with 20% (vol/vol) fetal bovine serum (FBS) (HyClone), penicillin (100 U/mL), and streptomycin (100 µg/mL). HCT 116 cells were seeded onto glass coverslips in 24-well plates (4 × 10^5^ cells per well) for one day to reach the confluence. Monolayers were infected with purified spore suspensions (MOI: 100) in 200 µL of culture medium without FBS and the culture plates incubated at 37 °C for 3 h, 5% CO_2_ [58]. After incubation, the unbound spores were removed by washing three times with DPBS, HCT 116 cells were then lysed with 100 µL 0.5% Triton X-100 for 30 min at 37 °C, plated onto an LB agar plate and incubated under aerobic conditions at 37 °C overnight. After 24 h of incubation, the number of bacterial colonies were counted, and the adherence represented as number of spores/cell. The experiment was conducted in triplicate.

## Figures and Tables

**Figure 1 ijms-23-14946-f001:**
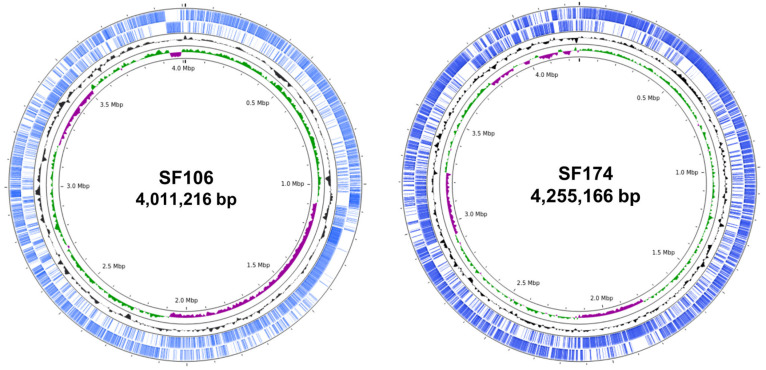
The circles represent from inside to outside: circle 1, DNA base position (bp); circle 2 and 3, GC content and GC skew; circle 4 protein-coding regions transcribed on the plus strand (clockwise); circle 5, protein-coding regions transcribed on the minus strand (anticlockwise). The genome plots were generated using Proksee.

**Figure 2 ijms-23-14946-f002:**
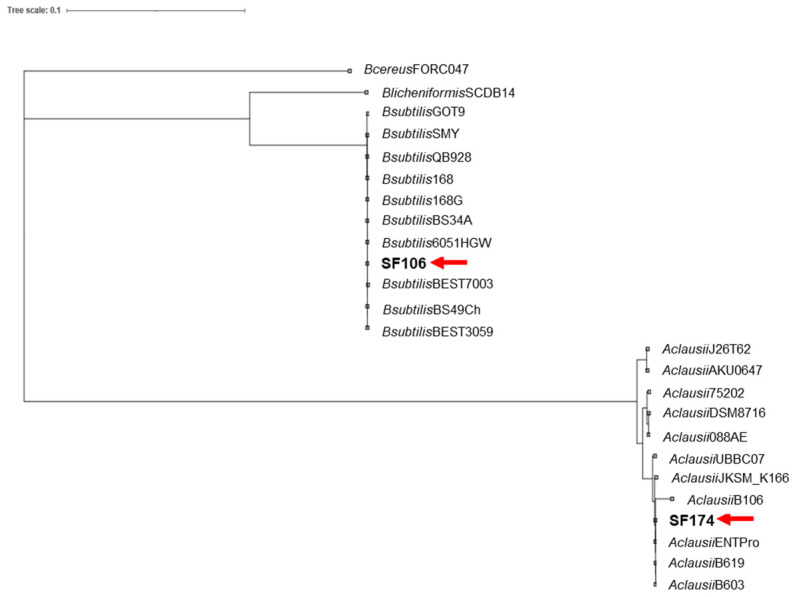
Phylogenetic tree of genome sequences of SF106 and SF174 (here indicated by the red arrows) and several B. subtilis and A. clausii species from NCBI, constructed using the tool GTDB. Strain names are reported. B. cereus and B. licheniformis were used as outgroups.

**Figure 3 ijms-23-14946-f003:**
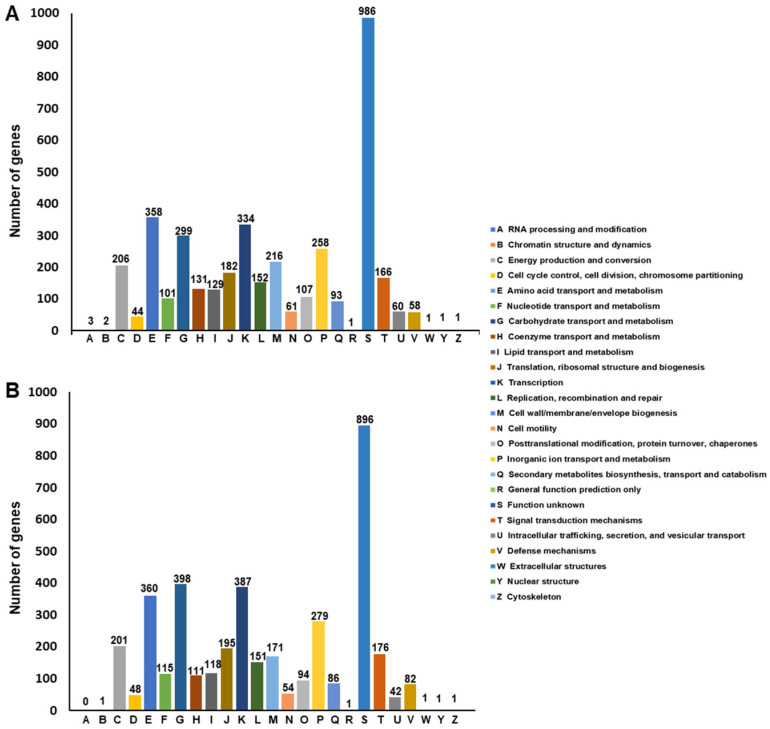
The COG function annotation of SF106 (**A**) and SF174 (**B**). The total number of proteins assigned to each category is indicated.

**Figure 4 ijms-23-14946-f004:**
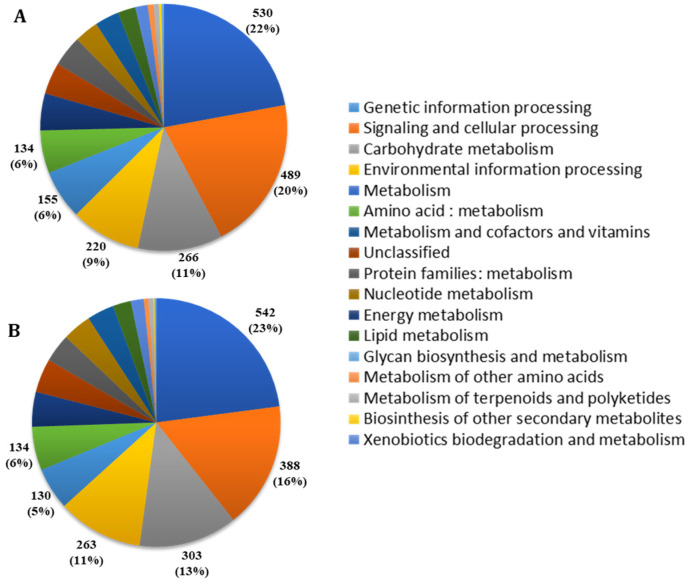
Genome annotation of SF106 (**A**) and SF174 (**B**) by BlastKOALA Kyoto Encyclopedia of Genes and Genomes (Kegg). The total number and the relative percentage of proteins assigned to the most abundant categories are indicated.

**Figure 5 ijms-23-14946-f005:**
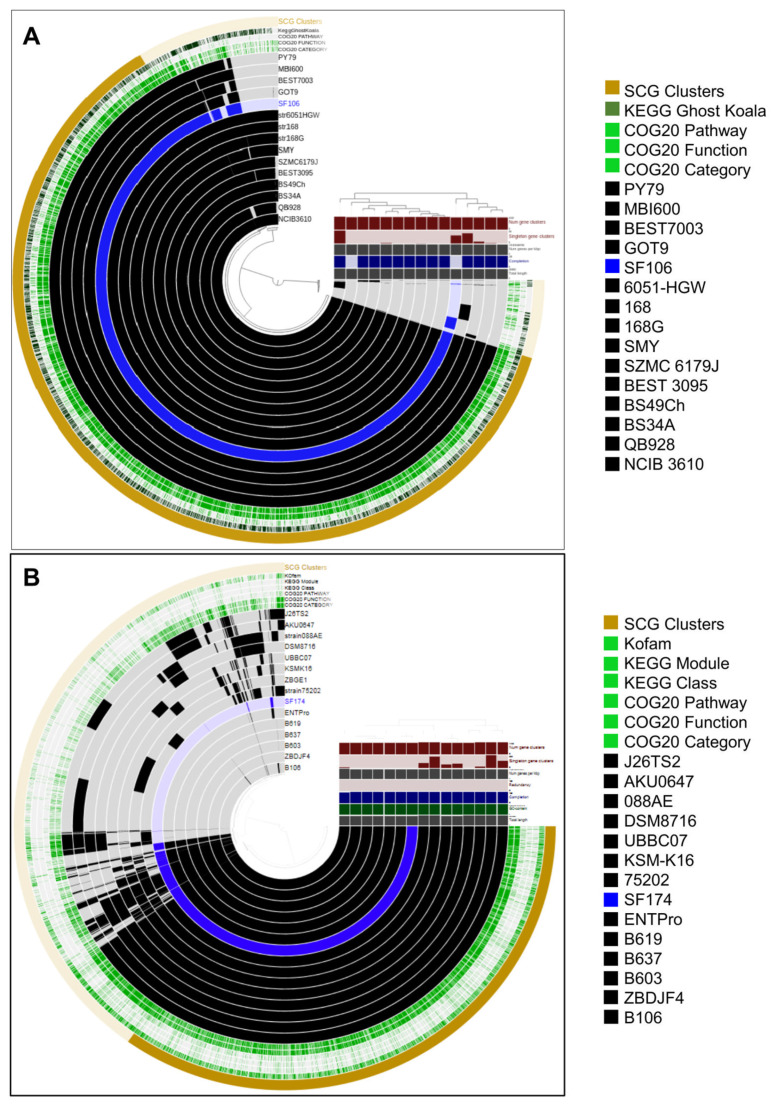
Anvi’o representation of the pangenome of the 15 *B. subtilis* (**A**) and 15 *A. clausii* genomes (**B**) generated with the items order in presence/absence (D: Euclidean; L: Ward). Each blue or black layer represents the genome of a strain, in blue SF106 (**A**) and SF174 (**B**) and all others in black. The additional layers indicate the single-copy gene (SCG) clusters (present once in each genome), the KEGG Ghost Koala or Kofam (KEGG Orthology and Links Annotation), KEGG module or class, the COG pathway, function and category.

**Figure 6 ijms-23-14946-f006:**
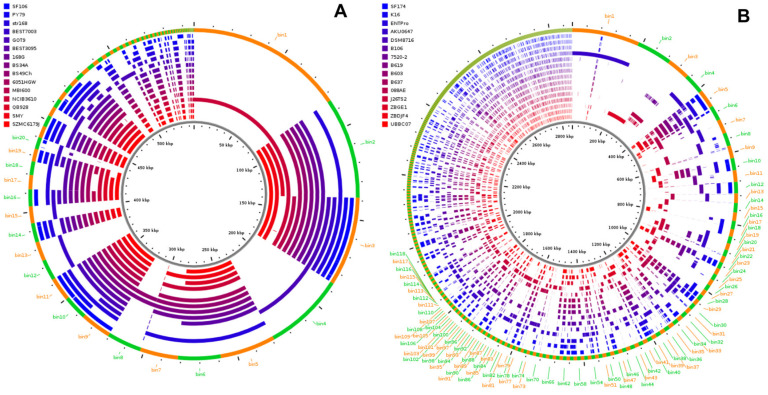
Distribution of the accessory genes of the 15 *B. subtilis* (**A**) and 15 *A. clausii* (**B**) genomes identified by the pangenomic analysis. The outer ring shows ClustAGE bins at least 200 bp in size ordered clockwise from largest to smallest with alternating orange and green colors to indicate bin borders. The ruler in the center of the figure indicates the cumulative size of the accessory genome bin representatives in kilobases. Figure generated using ClustAGE Plot utility available at http://vfsmspineagent.fsm.northwestern.edu/cgi-bin/clustage_plot.cgi, accessed on 1 April 2022, (version 0.8).

**Figure 7 ijms-23-14946-f007:**
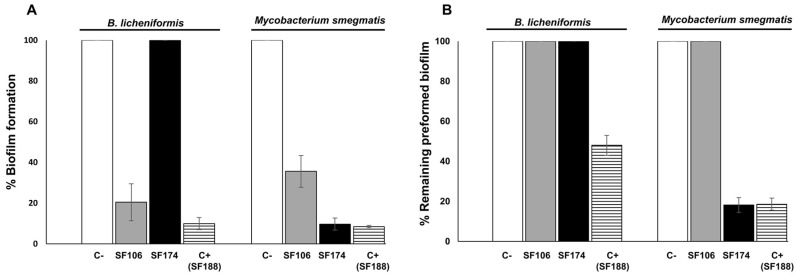
Effects of cell-free supernatants of SF106 and SF174 in preventing biofilm formation (**A**) and in disassembling pre-formed biofilms (**B**) of the indicated target bacteria. Biofilm formation and dis-assembly were assayed without any cell-free supernatant (C-) and with the cell-free supernatant of strain SF188 (C+), a *B. pumilus* strain [22], here used as positive control.

**Figure 8 ijms-23-14946-f008:**
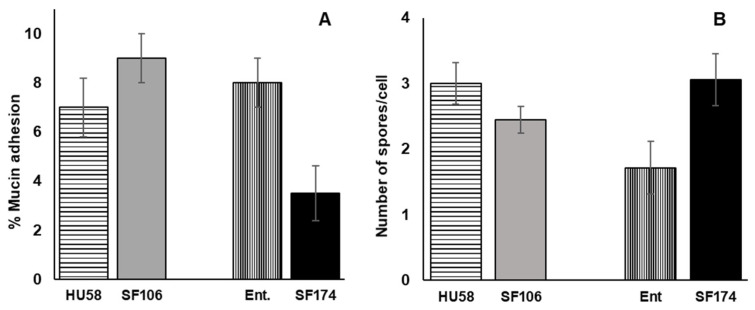
(**A**). Adhesion of *B. subtilis* (SF106 and a commercial probiotic of the same species HU58) and *A. clausii* (SF174 and a commercial probiotic of the same species, ENTPro) spores to human mucin (**A**) and to HCT116 cell line (**B**). The data represent the mean of three independent experiments, and the error bars are the standard error of the mean.

**Table 1 ijms-23-14946-t001:** General features of the SF106 and SF174 genomes.

	SF106	SF174
Size (bp)	4,011,216	4,255,166
Number of contigs	11	40
Average GC percentage	43.75	44.67
Number of predicted ORFs	3992	4225
Number of predicted rRNA genes	8	12
Number of predicted tRNA genes	63	75

**Table 2 ijms-23-14946-t002:** Number of putative genes for the CAZymes categories in SF106, SF174 and selected probiotic strains of the respective species.

Strain	GH	GT	PL	CE	CBM	Total
**SF106**	**53**	**39**	**7**	**14**	**22**	**135**
*B. subtilis* Natto BEST195	55	38	5	13	34	145
**SF174**	**54**	**28**	**4**	**12**	**13**	**111**
*A. clausii* EntPro	54	30	4	12	13	113

**Table 3 ijms-23-14946-t003:** Number of putative genes for families of the esterases and glycoside hydrolases CAZymes categories in SF106 and SF174.

Strain	GH2	GH3	GH18	CE4	CE9
SF106	0	1	4	6	1
SF174	1	2	2	6	1

**Table 4 ijms-23-14946-t004:** Antimicrobial activity of SF106 and SF174 against a panel of target bacteria.

Target Bacteria	SF106	SF174
*Pseudomonas fluorescens* ATCC 13525	-	-
*Salmonella enterica typhi* ATCC 14028	-	-
*Escherichia coli* ATCC 25922	-	-
*Shigella sonnei* ATCC 25931	-	-
*Bacillus subtilis* NCBI 3610	-	-
*Bacillus cereus* GC105	+^s/c^	-
*Bacillus licheniformis* SG277	-	-
*Listeria monocytogenes* ATCC7644	+^s/c^	+^c^
*Lactobacillus rhamnosus* GG	-	-
*Mycobacterium smegmatis* mc^2^ 155	-	-
*Staphylococcus aureus* ATCC 6538	+^c^	-
*Staphylococcus aureus* ATCC 25923	+^c^	-
*Staphylococcus aureus* ATCC 43300 MRSA	+^c^	-
*Staphylococcus aureus* ST398 MDR	+^c^	-
*Staphylococcus aureus* 392 MDR	+^c^	-
*Candida albicans*	+	+

s/c: activity observed with cell-free supernatant and by cell–cell contact; c: activity observed by cell–cell contact.

**Table 5 ijms-23-14946-t005:** Minimal Inhibitory Concentration (MIC) of selected antibiotics against SF106 or SF174.

Antibiotic	*B. subtilis* SF106 ^a^	*A. clausi* SF174 ^a^	*Bacillus* EFSA Breakpoint ^a^
Tetracycline	4.00	7.50	8
Vancomycin	2.00	0.50	4
Streptomycin	32.00	16.00	8
Clindamycin	1.19	16.00	4
Gentamycin	0.44	0.19	4
Erythromycin	0.07	16.00	4
Kanamycin	0.75	7.50	8
Chloramphenicol	1.00	24.00	8

^a^ Values are µg/mL.

## Data Availability

Not applicable.

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
