# Peer review of "Comparative Genomics and Physiological Characterization of Two Aerobic Spore Formers Isolated from Human Ileal Samples"

_ijms, 2022, doi:10.3390/ijms232314946_

Round 1
Reviewer 1 Report
The merit of this paper heavily lies in the justification of studying the two particular strains. However, few explanations were given in the Introduction part. The authors are therefore suggested to provide additional reasons to explain why looking at these two specific strains and to indicate the significance this study will bring.
The biofilm inhibition function of the two strains will be more clearly illustrated by including a "positive control", i.e. other bacteria known to inhibit biofilm formation strongly. In fact, that's what was done in the AMR studies.
The data were presented clearly, and the manuscript was well organised.
Reviewer 2 Report
In the manuscript entitled ''Comparative genomics and physiological characterization of two aerobic spore formers isolated from human ileal samples'', the authors present a genomic and physiological characterization of two aerobic spore former strains previously isolated from 19 ileal biopsies of healthy human volunteers. By using genome sequence and phylogenetic analysis, they showed that the two strains SF106 and SF174 belong to the B. subtilis and Alkalihalobacillus clausii (formerly Bacillus clausii) species, respectively, and are unable to produce toxins or other 21 metabolites with cytotoxic activity against cultured human cells, efficiently bind mucin and human epithelial cells in vitro and produce molecules with antimicrobial and antibiofilm activities. The data presented suggest that the two strains do not pose safety concerns and have beneficial potentials for probiotic treatment. . This is a well witten paper and the introduction, results, discussion and materials and method sections are presented with clarity. The methodology used is adequate. The references cited are appropriate.
Minor comments:
Please, correct the species names into the italics form throughout the manuscript. The in vivo and in vitro should also be in the italics form, as well.
